# Agronomic Factors Influencing Fall Armyworm (*Spodoptera frugiperda*) Infestation and Damage and Its Co-Occurrence with Stemborers in Maize Cropping Systems in Kenya

**DOI:** 10.3390/insects13030266

**Published:** 2022-03-07

**Authors:** Daniel Munyao Mutyambai, Saliou Niassy, Paul-André Calatayud, Sevgan Subramanian

**Affiliations:** 1International Centre of Insect Physiology and Ecology (ICIPE), Nairobi P.O. Box 30772-00100, Kenya; sniassy@icipe.org (S.N.); pcalatayud@icipe.org (P.-A.C.); ssubramania@icipe.org (S.S.); 2UMR Évolution, Génomes, Comportement et Écologie, IRD, Université Paris-Saclay, CNRS, 91198 Gif-sur-Yvette, France

**Keywords:** agro-ecology, agronomic practices, fall armyworm, Kenya

## Abstract

**Simple Summary:**

Fall armyworm (FAW), an invasive pest of maize and other cultivated crops, has been established in Kenya since 2016. It is a serious threat to maize production and poses a challenge to food and nutrition security. Little is known about its co-occurrence with resident stemborers, relative infestation and damage and how agronomic factors influence its infestation and damage in maize cropping systems across different agro-ecological zones. Maize fields were surveyed across three agro-ecological zones in Kenya. Fall armyworm was found across the three agro-ecological zones and it dominated resident stemborers in maize cropping systems. Its infestations and damage were highest at coastal lowlands compared to mid-altitude and high-altitude lands. Maize grown under mixed cropping systems, with rainfed production and weeded frequently had low infestation and damage compared to those grown under monoculture, with irrigation and no weeding, respectively. Young vegetative maize plants were more infested and damaged compared to mature plants. Different maize varieties were found to have different infestation and damage levels. These results demonstrate dominance of FAW over resident stemborers and that agronomic practices play a role in influencing FAW infestation and damage in maize cropping systems, which need to be considered when designing sustainable pest management solutions.

**Abstract:**

Fall armyworm (FAW), *Spodoptera frugiperda* J.E Smith, (Lepidoptera: Noctuidae) is a serious invasive pest of maize that has been established in Kenya since 2016. Little is known about its co-occurrence with resident stemborers, relative infestation and damage and how agronomic factors influence its infestation and damage in maize cropping systems across different agro-ecological zones. This study assessed FAW co-occurrence with resident stemborers, relative infestation and damage across three agro-ecological zones, and the effects of different agronomic practices on its infestation and damage in maize cropping systems in Kenya. A total of 180 maize farms were surveyed across three different agro-ecological zones. FAW infestation and damage was highest in lowlands compared to mid-altitude and high-altitude lands. Its population (eggs and larvae) dominated that of resident stemborers in maize fields. Maize grown under mixed cropping systems, with rainfed production and weeded frequently had low infestation and damage compared to those grown under monoculture, with irrigation and no weeding, respectively. Young vegetative maize plants were more infested and damaged compared to mature plants. Different maize varieties were found to have different infestation and damage levels with Pioneer having the least damage. These results demonstrate that agronomic practices play a role in influencing FAW infestation and damage in maize cropping systems. Further, the population of FAW is dominating that of stemborers in maize cropping systems in Kenya, four years after its invasion. Thus, agronomic practices need to be considered while designing sustainable agro-ecological-based management solutions for resource-constrained smallholder farmers.

## 1. Introduction

Globally, maize, *Zea mays* L. (Poaceae), is the third most important cash and food crop after rice and wheat. It is an important source of food and nutritional security for millions of people in sub-Saharan Africa, accounting for 73% of calorific intake [1]. In Kenya and most of Africa, maize is predominantly produced by smallholder farmers under mixed crop–livestock farming systems [2,3] and is grown across a wide range of agro-ecological zones, from wet to hot semi-arid lands and in different soil types compared to other cereal crops [1]. Maize production is constrained by drought, diseases and several pests, including lepidopteran cereal stemborers, such as the native *Busseola fusca* (Fuller) (Noctuidae), *Sesamia calamistis* (Hampson) (Noctuidae) and invasive *Chilo partellus* (Swinhoe) (Crambidae), which can cause tremendous yield loss. For example, in Kenya, yield loss due to *C. partellus* is reported to be 80% in sorghum crop [4]. The recent invasion by the fall armyworm (FAW), *Spodoptera frugiperda* J.E Smith, (Lepidoptera: Noctuidae) in Kenya further threatens maize production and poses an additional threat to food and nutrition security for millions of people [5].

Fall armyworm is a voracious pest native to tropical and subtropical America. It was first detected in Kenya in 2016 and has been causing extensive damage to crops [6]. It is a polyphagous pest that is known to attack 353 plant and crop species belonging to 76 plant families, mainly Poaceae (106 taxa), Asteraceae and Fabaceae (31 taxa each) in its native range [7]. The destructive stage of FAW is the larval stage, whose caterpillars feed on young leaf whorls, ears and tassels, inflicting substantial damage to maize crops and causing high grain yield loss [8]. Late larval instars can cut through the entire base of the young maize seedling, killing the whole plant [9].

Vegetative maize crops can recover from FAW damage, especially if the crop is growing rapidly. However, impacts of heavy damage levels and late detections of infestations may lead to irreversible damage [10]. Previous studies have reported the maize leaf damage score by FAW in Kenya to range from 3.2 to 5.3 on the Davis damage scale [11]. Fall armyworm cause similar damages and occupy the same ecological niches as lepidopteran stemborers, such as *B. fusca*, *S. calamistis* and *C. partellus* [12]. These lepidopterans are interacting with and amongst each other, especially at the larval stage in the utilization of crop resources, being mainly maize, which is their preferred host plant [12,13]. Given that these herbivore pests have similar feeding guild, host plant and phenological characteristics [14,15], competitive interspecific interactions are likely to occur. These multi-faceted interactions will persist as FAW and stemborers continue to co-occur on several widely cultivated plants. Such interactions are likely to influence community structure of these lepidopteran herbivores in areas where they co-exist. Recently, field studies in maize and sorghum cropping systems in Uganda revealed that the invasion of FAW has caused the decline of stemborer incidences and the displacement from the maize crop, as their preferred host plant, to sorghum [16]. Assessing interactions and co-occurrence across different agro-ecologies is critical to inform the best pest management strategies to be deployed.

Comprehensive studies, assessing the role of agronomic factors on the infestation level and yield loss in maize due to FAW in Kenya since its invasion of the country, are lacking. Kumela et al. [17] reported an infestation level of 47% and yield reduction of 1381 kg/ha in five sub-counties that were assessed. They also established that farmers’ knowledge and perceptions of FAW management varied across countries (Ethiopia and Kenya), pinpointing the need to develop management strategies for this pest based on the local farmers’ needs and priorities. De Groote et al. [6] reported that since the FAW arrival in Kenya in 2016, by 2018, it had spread to all major maize-growing zones and affected 86% of the farmers in the country. They reported yield loss due to this pest to be 33%, varying across different agro-ecological zones. These studies pointed out the huge impact that FAW is causing in maize production in Kenya. However, both studies relied mainly on community-based surveys through interviews of farmers in focused group discussion. They lacked systematic field sampling assessing different agronomic factors that could contribute to yield losses associated with FAW and its interactions with other similar pests, such as stemborers, which infest and cause similar damage to maize.

Due to the high losses caused by FAW, after its first detection the use of insecticides to control the pest was the main focus, and many governments in Africa procured high volumes of insecticides to control the pest. Continuous use of these chemical insecticides poses adverse risks to human and environmental health, including the loss of pests’ natural enemies, pollinators and biodiversity. Additionally, costs associated with these chemicals make them unaffordable by most smallholder farmers, necessitating a search for alternative sustainable options. A range of alternative agro-ecological-based control measures, including maize–legume intercropping and climate-smart push–pull companion cropping, have been demonstrated to reduce FAW damage on crops [9,18,19]. These management strategies rely on locally available biodiversity and local knowledge combined with farmer practices to manipulate the pest through habitat diversification. Such management strategies modify biophysical environment of farms, including soil health and consequently affect pest behavior through different means such as behavioral modulating cues, visual ability and dispersal abilities. These changed biophysical characteristics of the farm eventually affect pest colonization, establishment and eventual damage of the crop.

Deployment and fine-tuning of such sustainable FAW control options for optimal performance requires an understanding of current smallholder agronomic management strategies and livelihood factors and how these different factors are likely to interact, to inform on best ways to adopt them through a farming system approach and taking into consideration potential trade-offs [20,21]. Because of diverse, biophysical and socio-economic contexts in areas invaded by FAW, there is a need to assess the agronomic factors to understand their distribution, influence on FAW infestations, damage and interactions with potential agro-ecological sustainable solutions for FAW management. Thus, the objectives of this study were to: (1) assess FAW infestation and damage levels in smallholder maize cropping systems across different agro-ecological zones in Kenya; (2) determine agronomic factors influencing FAW infestation and damage in smallholder maize cropping systems in Kenya; and (3) assess the magnitude of the co-occurrence of FAW and stemborers in smallholder maize cropping systems in Kenya.

## 2. Materials and Methods

### 2.1. Study Sites

The study was conducted in 18 major maize-growing counties in Kenya (Figure 1, Appendix A). These counties are characterized by a suitable environment for FAW survival, including abundant maize crops, especially during the rainfed growing season, as well as the dry season cultivation of maize and other crops in irrigation schemes and on riverbanks, allowing the pest to persist throughout the year. For this study, altitude was considered a key factor in categorizing agro-ecological zones into three main categories: lowland, midland and highland agro-ecological zones. For each agro-ecological zone, representative counties for the study were selected based on the intensity of maize farming. For the lowlands, the counties of Kilifi and Kwale were selected. These counties are located along the Kenya coastal strip, with altitudes ranging from sea level to about 900 m above sea level and lie within the Kenya agroclimatic zones IV and V [22]. They are characterized by diverse crops and cropping systems [23]. The main crops are maize (*Z. mays*), cassava (*Manihot esculenta* Crantz), coconut palms (*Cocos nucifera* L.), legumes and a variety of vegetables. Mean annual rainfall of the coastal lowlands is over 1200 mm, while mean maximum annual temperature ranges between 30–34 °C [23]. Sandy, luvisols and red volcanic clay soils are the major soil types [23]. For purposes of this study, the counties of Busia, Kitui, Kirinyaga, Taita Taveta and Siaya constituted the midland agro-ecological zone. The altitude of these counties ranged from 901–1200 m above sea level on average and are characterized by bimodal rainfall with an average annual precipitation of 500–1000 mm. The mean maximum annual temperature ranges from 17.4 to 24.5 °C, and these counties lie within the Kenya agroclimatic zone IV [22]. Maize (Z. *mays*) dominates as the main crop. Other crops grown include sorghum (*Sorghum bicolor* L.), pearl millet [*Pennisetum glaucum* (L.) R.Br], legumes, sugarcane (*Saccharum officinarum* L.) and various vegetables, including traditional African vegetables. Soils are mainly nitisols, consisting of well-weathered clay soils [24]. Bomet, Bungoma, Embu, Homabay, Kakamega, Kisii, Migori, Nakuru, Narok, Trans-Nzoia and Vihiga counties constituted the highland agro-ecological zone. Agriculturally, this is the most productive agro-ecological zone in Kenya, with altitudes from 1201 m above sea level going up to 2700 m above sea level, annual precipitation of 950 to 1500 mm and mean maximum annual temperature range of 16–24 °C [25]. These counties lie within Kenya agroclimatic zones II and III and are dominated by crop–livestock farming systems. Maize (*Z. mays*), coffee (*Coffea* species), tea [*Camellia sinensis* (L.) Kuntze], sugarcane (*S. officinarum*), wheat (*Triticum aestivum* L.) and various vegetables are the main crops grown. Soils mainly consist of vertisols and nitisols [26].

### 2.2. Research Design

The study comprised a systematic field sampling of different maize cropping systems and a questionnaire survey involving 180 smallholder maize growers.

#### 2.2.1. Field Selection

In each county, 10 farms were selected for the study. With the help of local agricultural extension officers, one farm per village where maize was actively growing was randomly picked for survey. To avoid biases of concentrating within one locality and ensure that the entire county is surveyed, the distance from one village of study to the next was approximately 15 km apart. The minimum size of the surveyed farm was 0.25 ha.

#### 2.2.2. Questionnaire Survey

To gather data on the socio-demographic, field characteristics and management practices, a total of 180 household heads of the selected farms were individually interviewed using a standard questionnaire (Appendix B). Permission to assess FAW and stemborer abundance, infestation and damage levels in maize crops in the farms was sought from each grower. The questionnaire comprised both open- and close-ended questions. Data captured in the questionnaire included farm characteristics (farm size, soil type, hedgerow, previous crop, irrigated or rainfed); cropping system (monoculture, row intercrop, mixed/broadcast cropping without any specific pattern, border cropping, strip cropping, push–pull companion cropping and duration of the cropping system on the farm); cropping pattern (fallow, rotation or continuous); tillage (conventional, conservation or zero) and mode of tillage (tractor, ox-plough or handhoe); crop characteristics (growth stage using V-notation); maize varieties planted (open pollinated, hybrid or recycled seeds); watering system—either rainfed or irrigation; fertilization (type of fertilizer—either synthetic including its name, manure or compost and the quantity applied); crop protection regimes, including insecticide use, name of insecticide, date of application, number of applications and dosage; weeding regimes, including number of weeding operations, use and name of herbicide(s), number of applications and dosage; other crop protective remedies applied and the farmer’s perception of FAW in relation to stemborers with regard to the damage caused to maize plants. Open-ended questions were used for continuous variables that included the age and education level of the smallholder farmer, farming experience and application rates of fertilizers, herbicides and insecticides. Gender of the interviewed grower was also captured as a categorical variable. For this study, conventional tillage was defined as any land preparation practice that involved the use of tractors and ox-plough to prepare land by burying plant residues and exposing the soil. Conservation tillage was categorized as any land preparation that involved minimum disturbance of the soil using handhoes and other means, where plant residue materials were left covering the soil. Zero tillage or no-till was described as any land preparation practice that had zero soil disturbances, except in making the holes for planting of the seeds. The questionnaire was prepared by the authors and field technicians using information from field observations and literature. It was pilot-tested on eight respondents randomly selected from smallholder maize farmers and modified before being administered. Questionnaires were administered to farmers through face-to-face interviews by experienced field technicians and researchers who had received prior training.

### 2.3. Data Collection

Data on FAW and stemborer abundance, infestation and damage levels in maize crop were collected through systematic sampling of 180 selected farms after interviewing the farmer and getting permission to sample the farm. Sampling started in the dry month of August 2019, extending through the October-December short rainy season and March–May 2020 long rainy season.

#### 2.3.1. Fall Armyworm and Stemborer Abundance and Maize Plant Infestation Levels

Fifty plants in each farm were screened for FAW and stemborer infestation either as the active presence of eggs and larvae or damage caused by these pests. *Busseola fusca*, *S. calamistis* and *C. partellus* were the stemborers of interest, as they are the most abundant stemborer species in maize crops in Kenya. Maize plants for infestation and damage screening were randomly selected using the ‘W pattern approach’ [27]. Maize plants in the outer two rows of each farm were not selected for screening. Scouting for eggs and larvae entailed first observing uncovered eggs and larvae on maize parts without causing major plant vibration disturbance, because larvae are sensitive to vibrations and quickly drop to the ground as a means of escaping from enemies [28]. Afterwards, leaves, whorls, ears and stems were thoroughly checked for any eggs and larvae that could be hiding inside the plant parts. A sharp scalpel blade was used to open the whorl region and stem where tunneling holes were found to collect the larvae. The number of egg batches and larvae of fall armyworm and stemborers were recorded separately. Fall armyworm and stemborer abundance was recorded as the total number of eggs and larvae in the farm (50 plants). Fall armyworm infestation level (%) was calculated as the number of plants with eggs/larvae and/or exhibiting symptoms of pest damage divided by the total number of plants screened (50) multiplied by 100.

#### 2.3.2. Level of Maize Plant Damage

The feeding damage score on the leaves of the randomly selected 50 plants per farm was assessed and rated using the Davis scale from 0 to 9, where 0 represents no damage, 1–4 represents low damage, 5–7 represents medium damage and 8–9 represents high damage for the vegetative stage [29]. The average damage score of each of the 50 plants per farm was recorded by taking the average of all the sampled plants, both infested and clean ones, which were recorded as 0.

### 2.4. Statistical Analysis

Data on the maize field and crop characteristics, as well as management practices, including the cropping system and pattern, tillage, fertilization, weeding and crop protection strategies, were reported using descriptive statistics. Socio-demographic data on farmer gender and level of education was analyzed using the Chi-square test. F-test was used to analyze data on the age and farming experience of interviewed smallholder maize farmers. The variability of the proportion of maize plants with FAW and stemborer infestation in each maize farm was analyzed using a generalized linear model (GLM). Being proportion data, a logit distribution was used to analyze maize infestation in different maize cropping systems [30]. Monoculture, arenosols (sandy) soils, maize as a previous crop, rainfed as a production system, continuous cropping as cropping pattern, conventional tillage, vegetative crop stage, HB500 series as maize variety, no fertilizer application, weeding and no insecticide use were used as reference variables. A multivariate analysis of variance was used to analyze maize damage data. Multiple pairwise comparisons of means were performed using Tukey’s multiple comparisons tests in the R package, ‘lsmeans’ [31]. Unpaired Students’ *t*-test was used to analyze the number of eggs and larvae of FAW and stemborers in the maize fields. As the variables under study did not occur at the same frequency across the agro-ecological zones, data analysis for factors affecting FAW infestation and maize damage were analyzed at two levels; disaggregated data for each agro-ecological zone and then aggregated data for the whole dataset. All the statistical analyses were carried out in R v.4.0.3 [Core Team, 2020] and α set at 0.05.

## 3. Results

### 3.1. Socio-Demographic Characteristics of Respondents

The majority of the respondents were male (60.6, Table 1). The age of the respondents ranged from 23 to 80 years with an average of 47.9 years. The age of farmers across the agro-ecological zones did not show any significance difference (Table 1). However, it varied widely with most farmers (66.7%) being middle-aged (between 35–60 years), 18.3% being younger than 35 years and only 15.0% being older than 60 years. The average farming experience was 14.5 years, and it did not significantly differ across the agro-ecological zones (Table 1). A total of 48.3% of the farmers had attained primary level education (8 years of basic education), over a third (31.7%) had attained secondary education (12 years of basic education) and 11.7% had a tertiary level education (post-secondary level education). Only 8.3% of the respondents had not gone through any formal education. The education level did not differ across the agro-ecological zones (Table 1).

### 3.2. General Characteristics of the Maize Farms

General characteristics of surveyed maize fields are summarized in Table 2. The average size of surveyed maize farms was 0.6 ha. Most of the maize plants were at the vegetative stage (V4 to V9 stages) when they were scouted for FAW and stemborer infestation and damage. A mixed cropping system of various crop combinations was the most common cropping system across the surveyed maize fields as opposed to maize monocultures (59.2% and 40.8%, respectively). Continuous cropping and crop rotation with different crops were the common cropping patterns practiced in the smallholder maize fields surveyed (41.7% and 39.4%, respectively), with fallow cropping pattern being the least practiced cropping pattern across the surveyed fields. Hybrid maize varieties were the predominant maize varieties grown by the smallholder farmers across the scouted maize fields. Conservation tillage (56.7%) was the common tillage practice across the scouted maize fields followed by conventional tillage (43.3%). No zero tillage was observed in the surveyed maize farms. Majority of the maize fields surveyed were under rainfed production (70.0%) compared to irrigation (30.0%). Most farmers applied commercial fertilizers (77.8%), with very few farmers using compost/manure in their maize fields (10.6%). Weeding frequency depended on the maize stage with most farms (54.4%) having been weeded once across the sampled maize fields. Herbicide use was very rare (1.7%), as most farms were manually weeded. Insecticide use was common across the sampled maize fields, although there was inconsistency on regular spraying patterns, with the majority of farmers who used insecticides having sprayed their maize fields only once (85.7%).

### 3.3. Fall Armyworm Infestation

The proportion of maize plants infested by FAW across the entire sampled farms was on average 50.7 ± 3.6%. The FAW infestation level differed significantly between the lowland and highland agro-ecological zones sampled (*p* = 0.01). It was highest at the lowlands and lowest at the highlands zone (Figure 2). Maize grown under different agronomic and management practices, such as cropping pattern, cropping system, fertilizer application, pesticide use, previous crop, production system, tillage and weeding demonstrated different infestation levels across the entire dataset, ranging from 27.0% to 100.0% (Table 3, Appendix A). Maize grown under different field characteristics, such as plot size and soil types, also demonstrated significantly different FAW infestation levels, ranging from 35.1% to 99.0% (Table 3, Appendix A). Crop characteristics, including crop growth stage and maize variety, also recorded significantly different FAW infestation levels, ranging from 28.3% to 52.8% (Table 3, Appendix A).

The influence of these factors on FAW infestation differed across the three agro-ecological zones (Table 4). Young vegetative plants were highly infested by FAW compared to tasseling and mature maize plants except at the lowlands, where there was no difference between infestation levels in the young vegetative and tasseling stage. Leaving land fallow and crop rotation of maize with legumes, potatoes and vegetables recorded low FAW infestation in maize compared to continuous maize cultivation in the same farm except in the midlands zone. Maize monocultures had higher FAW infestations compared to mixed cropping systems across the agro-ecological zones except maize-cassava intercrops at the midlands zone, mixed cropping with no specific pattern at lowlands and maize-vegetable intercrops at the highlands zone. For the lowlands, there was no difference in FAW infestation in fertilized fields, while in the midlands and highlands, fertilizer application seemed to increase FAW infestation, as indicated by positive Z values. Similarly, there was no difference in FAW infestation in lowlands fields with or without insecticide use. However, the positive Z value in the midlands pointed to an increase of FAW infestation with insecticide use. There was no difference in FAW infestation levels in different plots in the lowlands, while at the highlands, plot size recorded positive Z values. Fall armyworm infestation varied across different maize varieties, with significant higher infestation observed in HB500 series, DK3081 and open pollinated varieties in the lowlands. In the highland zone, HB500 series, open pollinated varieties and Duma43 also recorded higher infestation compared to other maize varieties. Maize grown under irrigation had higher infestation except in the midlands. Maize plants growing in different soil types recorded different FAW infestation levels in the lowlands and highland zones with arenosols (sandy) having higher infestation levels. Maize plants grown under conventional tillage had higher infestation compared to those under conservation tillage, except in the midlands zone. Maize plants that were weeded had lower FAW infestations compared to those that were not weeded in the highlands zone.

### 3.4. Fall Armyworm Damage

The average maize damage level for the entire dataset across different agro-ecologies was 4.4 ± 0.33 for the vegetative stage, which is a medium damage on the Davis scale. Higher maize damage was observed in the lowlands and midlands at 6.3 ± 0.38 and 6.1 ± 0.48, respectively, with the least damage level being observed in the highland zones at 2.5 ± 0.33 (Figure 3). Maize grown under different cropping systems, production systems, soil types and weeding regimes recorded significant different damage levels across the entire dataset (Table 5, Appendix A). Significant different damage levels were also observed in different maize varieties (Table 5, Appendix A). Mixed cropping systems, especially push–pull companion cropping, maize-legume intercropping and mixed cropping without any specific pattern, had lower damage levels compared to maize monocultures (1.1 ± 0.34, 2.6 ± 0.70 and 5.1 ± 0.38, respectively). Maize plants grown in farms with sandy (arenosols) soils had the highest damage level compared to maize grown in farms with other soil types (7.3 ± 0.30, 4.7 ± 0.42, 3.4 ± 0.73 and 2.2 ± 0.61 for arenosols, nitisols, vertisols and ferrasols, respectively). Maize plants grown under an irrigation production system had a significantly higher damage level than those grown under a rainfed production system (*p* = 0.038). Maize damage levels varied significantly among different maize varieties (*p* = 0.003), with the Pioneer variety having the least damage level and DH04 having the highest damage level among the maize varieties sampled during the field survey (Pioneer 1.8 ± 0.93; PH4 3.2 ± 0.84; DK777 3.3 ± 0.45; SC Duma 43 3.4 ± 1.70; HB6 series 4.7 ± 1.19; HB5 series 5.5 ± 1.60 and DH04 6.8 ± 0.17). Maize farms that were weeded more than once recorded significantly low maize damage compared to those that were not weeded (*p* = 0.007). Influence of these agronomic factors and field and crop characteristics were observed to be different across the different agro-ecological zones surveyed (Table 6). Of the total plants assessed, only eight plants had tassel, silks and ears damage and so these damages were not included in the data analysis.

### 3.5. Fall Armyworm and Stemborer Co-Occurrence and Abundance

Both FAW and stemborers were all found in most of the surveyed maize fields, with fall armyworm being the most abundant. A total of 1863 larvae and 130 egg batches were collected from the field surveys. Out of the 1863 larvae, 1620 were fall armyworm while 243 were stemborers. Of the 130 egg batches, 113 were fall armyworm while 17 were for stemborers. Across the agro-ecological zones, 704, 576 and 340, fall armyworm larvae were observed in the lowlands, midlands and highlands, respectively. For stemborers, 177, 60 and 6 larvae were found in the lowlands, midlands and highlands, respectively. For egg batches, 96, 13 and 4 fall armyworm egg batches were collected in the lowlands, midlands and highlands, respectively. For stemborers, 9, 8 and 1 egg batches were found in the lowlands, midlands and highlands, respectively. Out of the total number of maize plants where these lepidopteran pests were found, 61.4% had fall armyworm infestation alone, 6.0% of the plants had stemborers alone while 32.5% had both fall armyworm and stemborers co-existing (Figure 4). In maize plants where FAW and stemborers cohabitated, FAW was predominant compared to cereal stemborers (*C. partellus*, *B. fusca* and *S. calamistis*), both in the total number of egg batches (*p* = 0.001) and larvae (*p* < 0.001). The mean number of egg batches per 50 maize plants for the whole dataset was 1.1 ± 0.33 for FAW and 0.1 ± 0.04 for stemborers (Figure 5A). The average number of larvae found per 50 maize plants across the sampled fields was 17.4 ± 1.82 for FAW and 1.9 ± 0.45 for the stemborers (Figure 5B).

## 4. Discussion

Majority of farms surveyed were smallholdings, and mixed cropping systems dominated most of the farms. Fallow system was the least practiced cropping pattern due to limited land availability. Respondents of the sampled population had a relatively moderate literacy level, as only 8.3% had not gone through any formal education.

The levels of FAW infestation marked by the proportion of maize plants with active pest and/or damage ranged from 41.3% to 65.5%, and the damage score level of 4.4 ± 0.33 reported in this study are within the range of previous studies that have documented the infestation of FAW in sub-Saharan African smallholder farms ever since the invasion of FAW in the African continent, e.g., Kumela et al. [17] for FAW infestation in Ethiopia and Kenya; Baudron et al. [32] for FAW infestation in Zimbabwe. Fall armyworm infestation and damage in maize were found across all surveyed areas in Kenya, with the highest infestation and damage being found at the lowlands and followed by the mid-elevation midlands with the least infestation recorded in the high-altitude highlands. Although the counties surveyed were not equal, owing to the geography of Kenya, these results demonstrate rapid spread and establishment of FAW since its invasion in Kenya, which was first reported in western Kenya towards the end of 2015; by 2016, it was sighted in the agriculturally high-potential central Kenya highlands [6]. This level of infestation and damage of maize crops threatens food security and demonstrates that FAW has become a major constraint to maize production in the country. In the first years of its invasion in Kenya until 2017, FAW observations were more common in the highlands, followed by moist mid-altitude zones, with the dry midlands and lowlands having not observed FAW until 2017 [6]. Our results indicate that since the FAW arrival at the hot and wet coastal lowlands, it has established itself and is causing more damage on maize crops compared to the high-altitude highland areas. The coastal lowlands exhibit hot and wet weather conditions, which are favorable to the FAW as opposed to the cold high altitude highland areas. On the other hand, while the midlands experience hot weather conditions, the dry periods provide unfavorable conditions for FAW, except in the areas where crops are produced under irrigation during these dry periods. These results corroborate those observed by De Groote et al., 2020 [6], who demonstrated proportions of FAW infestations, damage and resultant yield loss in coastal lowlands and dry mid-altitude lands to have more than doubled, while those in the highlands had reduced from initial reports when FAW invaded Kenya. They attributed this to the availability of maize, as these areas have two maize growing seasons, namely short and long rain seasons, as opposed to only one maize-growing season in the highlands. The low temperature in the highlands could also have contributed to the reduced FAW infestation and damage, as FAW is known not to survive periods of extreme cold and those with mild cold and rainfall [33]. The lower temperature for FAW optimal growth has been suggested as 25 °C [34]. In some months, the Kenyan highlands experience lower temperatures below this optimal FAW growth temperature, at times even below 10 °C [35].

Our results demonstrate low maize infestation and damage by FAW in mixed cropping systems, including push–pull companion cropping, maize–legume intercrops either grown as row intercrops or in a broadcast manner without any specific pattern, including planting in the same hole with maize compared to maize grown as monocultures. Cropping systems and diversification have been demonstrated to reduce insect pests’ prevalence and damage by influencing behavior and population dynamics during colonization, establishment and population development phases of infestation [36,37]. Midega et al. [19] demonstrated the effectiveness of a climate-adapted push–pull companion cropping system against FAW in East African smallholder maize farms, although the mechanisms behind this observation were not outlined. Similar observations were made by Hailu et al. [18] on maize–legume intercrops in Uganda compared to maize monoculture. Guera et al. [38] demonstrated that maize push–pull cropping systems encompassing *Brachiaria* hybrid cv. Mulato II, *Panicum maximum* Jacq. cv. Mombasa and *Panicum maximum* cv. Tanzania as pull plants and *Dysphania ambrosioides* L., *Tagetes erecta* L. and *Crotalaria juncea* L. as push plants had lower levels of FAW infestations and higher yields compared to maize monocultures. Legume intercrops observed in the maize fields included common beans (*Phaseolus vulgaris* L.), cowpeas (*Vigna unguiculata* L.) and groundnuts (*Arachis hypogaea* L.). The presence of these legumes in mixed cropping systems enhance dispersal of FAW larvae away from main crop, thus reducing the damage inflicted on the companion host plants. The dispersal ability of FAW, especially the larval stages, could have been physically impeded where the intercrops are non-hosts. While dense host intercrops may provide a medium for the ballooning FAW larvae, the dense intercrops are also known to create a suitable habitat that harbors several natural enemies for insects, including both parasitoids and predators [39], which could have consequently reduced the FAW infestation and damage in maize-mixed cropping systems as opposed to maize monocultures. A mixture of crop types can also interfere with visual stimuli that attract insect pests to their suitable host crops, totally camouflaging the host crop, especially young plants in cropping systems, such as relay cropping systems [37].

Varying degrees of FAW damage were found among the different maize varieties surveyed, with the Pioneer variety having the least damage and DH04 having the highest damage level (Appendix A). The observed damage differences could be ascribed to different tolerance and defense mechanisms of different maize varieties against FAW, including maize morphology and phytochemistry [40,41,42]. African maize cultivars have been demonstrated to have differing levels of acceptance and feeding preference by FAW larvae [43], which could explain the observed varying degrees of damage in different maize varieties. Potential mechanisms of resistance against lepidopteran pests in maize have been elucidated, including the direct accumulation of phytotoxic protein molecules, such as maysin in silks, cellulose build-up at the cell wall, chlorogenic and aspartic acid [44], or phytochemicals, such as 2,4-dihydroxy-7-methoxy-2H-1,4-benzoxazin-3(4H)-one (DIMBOA) or flavonoids [45]. Indirectly, maize also defends itself against FAW by attracting natural enemies through volatile emission that attract parasitoids, such as the *Cotesia* species, for herbivore attack [42].

Maize under irrigation production systems was severely infested and damaged compared to those under rainfed production systems. This could be explained by the fact that irrigation takes place during the dry season when most of the vegetation around the irrigated farms are dry; thus, the growing maize act as the only host plants available, as opposed to the rainy season when there are plenty of host plants growing, including gramineous weeds, which act as a refuge for lepidopteran pests [33,46]. Rainfall has been demonstrated to influence the population dynamics of FAW by negatively influencing its survival [47]. The observed reduction in FAW infestation and damage in rainfed maize fields compared to the irrigated ones could also be due to the washing off effect of dispersing early instars and pupae in the soil by the rainwater, thus reducing the population of the pest on maize plants. Wyckhuys and O’Neil [48] reported low FAW infestations in high rainfall seasons and high infestations in dry years in Honduran smallholder maize farmers and ascribed this to the possibility of FAW larvae being washed off the whorl by the rainfall, as they do not burrow into the maize stems, unlike stemborers. Most of the irrigated fields had furrow irrigation, which does not confer the same forceful effect of water droplets compared to rainfall and other irrigation systems, such as sprinkler irrigation. Thus, while designing FAW management strategies, the production system in place needs to be considered for the selected strategy to achieve the desired results.

Disaggregating insecticide use data across the three agro-ecological zones demonstrated that there was no difference in the FAW maize damage between maize farms with or without insecticides in the lowlands zone. For midlands, higher FAW damage was observed in fields receiving insecticides treatment compared to those without insecticides. However, the fact that the coefficient for insecticide use was positive and of high absolute value indicates poor efficacy, or the farmers applied insecticides when high damages had already been inflicted by FAW, as most farmers apply insecticides after noticing plant damages. In the highlands, insecticide use resulted in lower maize damage, although there were low insecticides use in the highlands due to low levels of FAW infestation and damage. These results agree with what other researchers have found in sub-Saharan Africa smallholder farms in regard to insecticides use on FAW. For example, Baudron et al. [32] found higher FAW damage in farms receiving insecticides treatments in smallholder maize farms in Zimbabwe, while Kumela et al. [17] reported low efficacy of insecticides against FAW in Kenya. The infrequent application of these insecticides, as more than 85% of the surveyed farmers applied insecticides once or applied the wrong insecticides or wrong doses because most farmers reported using a mixture of insecticides, could contribute to this reported low effect on FAW. Used judiciously and timely applied in the FAW infestation phase, insecticides can lower damage inflicted by FAW larvae on maize. However, they pose adverse human and environmental health challenges, including negative effects on the non-target organisms, such as pollinators and FAW’s natural enemies [49].

Maize growing in sandy soils were more infested and severely damaged compared to those growing in other soils. This may be explained by the fact that soil nutrients and biology, including soil microbes, influence plant growth and phytochemistry, including defense metabolites through plant–soil feedbacks, which could have negatively influenced the feeding damage of the FAW larvae on the maize plants [50]. Soil type has been demonstrated to influence FAW pupal survival and adult emergence [47]. As FAW larvae pupate mainly in the soil, loose sandy soils provide a suitable environment for pupation, which can contribute to increasing the number of emerging moths and subsequent laying of eggs in the nearby maize plants. Further studies need to be conducted to unravel the exact mechanisms governing this observed phenomenon.

Land tillage practices were demonstrated to have negatively influenced the proportion of maize plants infested with FAW. Maize fields established through conservation tillage had fewer plants infested by FAW than those established through conventional tillage. However, land tillage did not affect the severity of damage inflicted by FAW on maize plants. The effect of land tillage practices on FAW infestation and damage has been reported in FAW’s native range in Florida and Mexico [51,52] and in its invaded areas of southern Africa in farms established through zero tillage [32]. These have been attributed to higher densities of generalist predators, including carabid beetles, spiders and ants in minimum-tillage farms [53]. Minimum disturbance to the soil provides organic mulch for the predators to hide and alternative prey for predators to increase their abundance [53]. Indeed, ants preying on FAW larvae were observed in maize fields that had ants nesting sites.

Frequent weeding tended to reduce both FAW infestation and damage in maize plants across different maize cropping systems. This can be explained by the fact that mechanical weeding, which was the common weeding practice observed, could have mechanically damaged the pupae, exposed the pupae in the soil to harsh weather conditions, including direct sun heat and predators by removing the shading effect of the weeds. Weeds are known to be FAW reservoirs; this has been demonstrated in Brazil [54]. Although these weeds may also host FAW’s natural enemies [55], they may also host other pests, including cereal stemborers with which FAW share the same ecological niche and may contribute to the damage inflicted on the maize plants [56,57]. If weeds, including the gramineous ones, which are closely related to maize, attract FAW, it will be advantageous to plant them around the maize field to attract FAW away from the maize, such as in the push–pull strategies that have been developed for stemborers [38]. This is one of the key principles utilized by push–pull companion cropping against stemborers by using Napier grass as a trap crop, as it is more attractive to the ovipositing females [39]. Though beyond the scope of this study, the determination of landscape structure in and around maize farms and its relationship with FAW abundance could further explain this observed phenomenon. Indeed, landscape complexity, including grasslands and their diversity near the farms, can determine the load of pests and parasitoids entering or exiting crop farms [58].

Field surveys recorded higher FAW numbers (eggs and larvae) than combined numbers of cereal stemborers, both native (*B. fusca* and *S. calamisitis*) and exotic *C. partellus* on maize plants. Although surveys conducted just at the beginning of FAW invasion in Kenya revealed a possible co-existence of FAW with stemborers [13,59], our results demonstrate that within an approximately four-year period since its invasion in Kenya, FAW has become the dominant lepidopteran pest species in local maize fields where they cohabitate with stemborers. Before FAW invasion, stemborers were the dominant lepidopteran pests in maize fields in Kenya, with *C. partellus* being dominant in low-altitude zones, while *B. fusca* dominated the high-altitude zones [60,61,62]. Recent laboratory studies have highlighted a higher FAW larval dispersal activity compared to stemborers [63] as well as predation among other lepidopteran pests [64]. Fall armyworm could be employing these competitive interaction mechanisms to dominate and displace stemborers in maize cropping systems. There is a need to characterize all stages of these herbivores, their niche differentiation at different maize phenological stages and their interaction mechanisms across each of the different agro-ecologies.

Due to its high preference for maize, FAW is likely to push stemborers to other host crops, such as sorghum and millet, increasing the pest burden in these crops. Such pest shifts have been reported in Uganda, where stemborer numbers have reduced in maize but increased in sorghum following FAW invasion [16]. Although in our data, we have grouped all stemborers together, future studies need to underscore which of the different stemborer species are impacted most by FAW invasion. Displacement effects caused by invasive lepidopteran insects have been reported in previous studies. For example, FAW has been reported to displace the resident common cutworm, *Spodoptera litura* in maize cropping systems in China, following the invasion of former in the country through interference competition [64]. Kfir [65] reported the displacement of *B. fusca* by the exotic invasive *C. partellus* in sub-Saharan Africa due to its competitive advantage consisting of ending diapause earlier. Similarly, Overholt et al. [66] found a displacement of indigenous *C. orichalcolciliellus* by invasive *C. partellus* to alternative grasses, which were not hosts of *C. partellus*. Therefore, the changing dynamics of FAW-stemborer populations in different cropping systems in Kenya and detailed interaction mechanisms both in the laboratory and field settings need further investigation to inform pest management strategies.

## 5. Conclusions

Results from this study have demonstrated that fall armyworm has established itself across all agro-ecological zones in Kenya, four years since its invasion in the country, and is more prevalent in coastal lowlands. Fall armyworm infestation and damage has been demonstrated to be higher and more severe in coastal lowlands followed by midlands and least in high altitude highlands. Agronomic practices, such as mixed cropping systems and weeding, as well as field characteristics, including soil type and crop characteristics like maize variety, were observed to negatively influence FAW infestation and damage in maize cropping systems. Fall armyworm has been demonstrated to dominate resident cereal stemborers in maize cropping systems where they cohabitate, which is an indication of possible displacement of cereal stemborers in maize due to its competitive advantages. These agronomic practices, as well as field and crop characteristics, need to be considered in designing sustainable FAW pest management strategies. The study recommends the incorporation of mixed cropping systems, weeding and the selection of appropriate maize varieties, as well as the evaluation and validation of other agronomic practices for sustainable FAW control, especially in smallholder farming systems.

## Figures and Tables

**Figure 1 insects-13-00266-f001:**
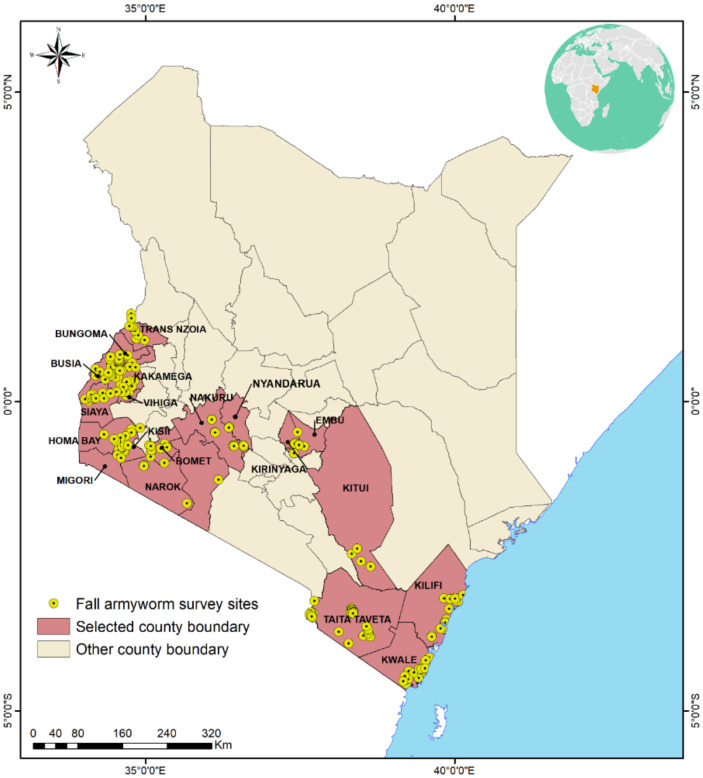
Map of Kenya showing all fall armyworm sites surveyed.

**Figure 2 insects-13-00266-f002:**
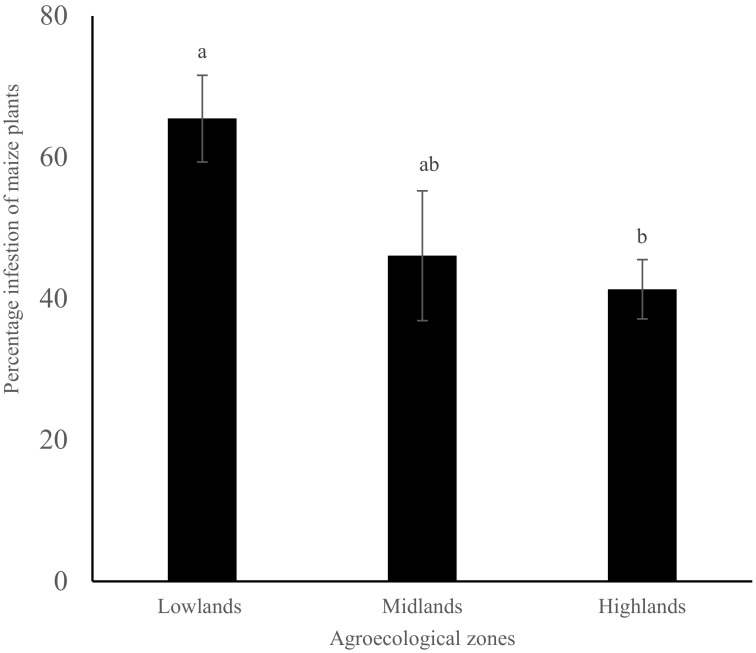
Percentage of maize plants infested (mean ± SE) by fall armyworm in different agro-ecological zones of Kenya. Bars capped with different letter differ significantly (Tukey’s studentized test: *p* < 0.05).

**Figure 3 insects-13-00266-f003:**
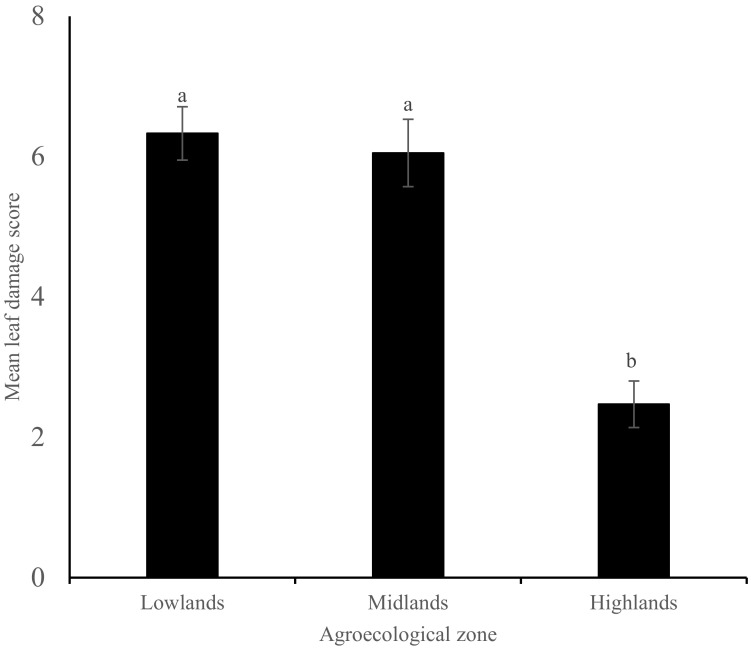
Mean fall armyworm damage score (± SE) on leaves of maize plants in three different agro-ecological zones of Kenya. Bars capped with different letter differ significantly (Tukey’s studentized test: *p* < 0.05).

**Figure 4 insects-13-00266-f004:**
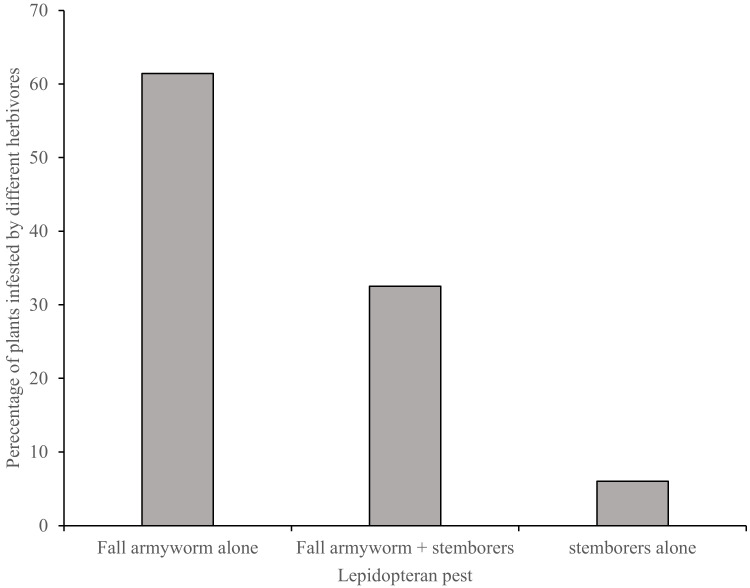
Percentage infestation of fall armyworm and stemborers, fall armyworm alone and stemborers alone in maize plants surveyed in different agro-ecological zones of Kenya.

**Figure 5 insects-13-00266-f005:**
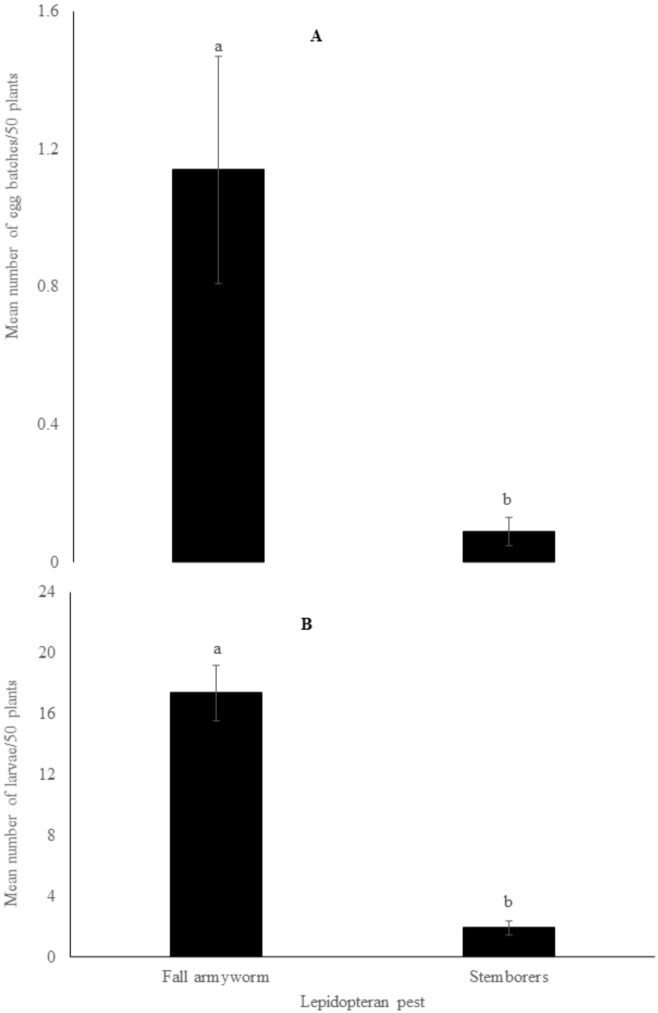
Mean number (mean ± SE) of egg batches (**A**) and larvae (**B**) of fall armyworm and stemborers per 50 maize plants observed during field survey in Kenya. Bars capped with different letter differ significantly (Tukey’s studentized range test: *p* < 0.05).

**Table 1 insects-13-00266-t001:** Socio-demographic characteristics of smallholder farmers interviewed during the survey for fall armyworm infestation and damage in three agro-ecological zones of Kenya in 2019/2020.

Variable	Agro-Ecological Zone			Mean N = 180	Significance	
	Coastal Lowlands	Midland	Highlands		χ^2^	*F*-Test
**Gender (%)**					0.14	
Male	75.0	60.0	46.7	60.6		
Female	25.0	40.0	53.3	39.4		
Age (years)	50.1	46.4	46.9	47.9		3.16 ns
Farming experience (years)	19.1	13.3	11.8	14.5		3.16 ns
Education level (%)					0.30	
Informal	5.0	10.0	10.0	8.3		
Primary	65.0	50.0	36.7	48.3		
Secondary	30.0	20.0	43.3	31.7		
Tertiary	0.0	20.0	10.0	11.7		

ns: denotes not statistically significant.

**Table 2 insects-13-00266-t002:** Maize field characteristics and crop management practices employed by smallholder farmers interviewed during the survey for fall armyworm infestation and damage in three agro-ecological zones of Kenya in 2019/2020.

Plot Variable	Lowlands Zone	Midlands Zone	Highlands Zone	Mean
Plot size	0.7 ± 0.07	0.5 ± 0.05	0.5 ± 0.04	0.6 ± 0.03
**Crop stage**				
Vegetative	61.1	61.8	83.3	68.8 ± 7.29
Tasselling	24.1	25.5	9.7	19.8 ± 5.03
Maturity	14.8	12.7	6.9	11.5 ± 2.35
**Cropping system**				
Monoculture	42.6	30.4	49.4	40.8 ± 0.05
Mixed cropping (No pattern) including pumpkins (MO)	11.1	10.9	14.8	12.3 ± 0.01
Maize-legume intercropping (ML)	14.8	43.5	23.5	27.2 ± 0.08
Push-pull (PP)	1.9	0.0	11.1	4.3 ± 0.03
Maize-agroforestry (MA)	5.6	0.0	0.0	1.9 ± 0.02
Maize-cassava intercrop (MC)	3.7	2.2	0.0	2.0 ± 0.01
Maize-legume-cassava intercrops (MLC)	16.7	8.7	0.0	8.5 ± 0.05
Maize-vegetable intercrops (MV)	3.7	4.4	1.2	3.1 ± 0.01
**Cropping pattern**				
Continuous cropping	55.0	20.0	50.0	41.7 ± 0.11
Rotation	15.0	60.0	43.3	39.4 ± 0.13
Fallow	30.0	20.0	6.7	18.9 ± 0.07
**Maize variety**				
OPVs	14.8	2.3	3.3	6.8 ± 0.04
Hybrids	46.3	77.3	90.0	71.2 ± 0.13
Recycled seeds	38.9	20.5	6.7	22.8 ± 0.13
**Tillage**				
Conventional	60.0	20.0	50.0	43.3 ± 0.12
Conservation	40.0	80.0	50.0	56.7 ± 0.12
Zero tillage	0.0	0.0	0.0	0.0 ± 0.00
**Production system**				
Irrigation	5.0	80.0	5.0	30.0 ± 25.00
Rainfed	95.0	20.0	95.0	70.0 ± 25.00
**Fertilizer application**				
Yes	70.0	80.0	83.3	77.8 ± 0.04
No	30.0	20.0	16.7	22.2 ± 0.04
**Manure/compost application**				
Yes	15.0	10.0	6.7	10.6 ± 0.02
No	85.0	90.0	93.3	89.4 ± 0.02
**Weeding frequency**				
Two or more	60.0	50.0	26.7	45.6 ± 0.10
Once	40.0	50.0	73.3	54.4 ± 0.10
**Herbicide use**				
Yes	5.0	0.0	0.0	1.7 ± 0.02
No	95.0	100.0	100.0	98.3 ± 0.02
**Insecticide use**				
Yes	47.2	42.9	10.0	33.3 ± 0.12
No	52.8	57.1	90.0	66.7 ± 0.12
**Insecticide frequency**				
Two or more	24.1	14.6	4.2	14.3 ± 0.06
Once	75.9	85.5	95.8	85.7 ± 0.06

**Table 3 insects-13-00266-t003:** Generalized linear models’ results for the aggregated data on maize field characteristics and crop management practices employed by smallholder farmers interviewed during the survey for fall armyworm infestation and damage in three agro-ecological zones of Kenya in 2019/2020.

Term	Incidence of Plants Infested by FAW			
	Estimate	Standard Error	Z Value	*p* Value
**Crop stage**				
Tasseling	−0.8	0.1	−12.3	<0.001
Mature	0.0	0.1	−0.4	0.692
**Cropping pattern**				
Fallow	0.5	0.1	3.9	<0.001
Rotation with legumes	0.4	0.1	2.8	0.006
Rotation with potatoes	0.5	0.2	2.5	0.012
Rotation with vegetables	−0.6	0.1	−5.0	<0.001
**Cropping system**				
Maize-cassava intercrop	−2.6	0.3	−9.3	<0.001
Maize-legume intercrop	−1.1	0.1	−13.0	<0.001
Maize-legume-cassava intercrop	−0.9	0.1	−9.0	<0.001
Mixed cropping (no pattern)	−1.6	0.1	−18.9	<0.001
Maize-vegetable intercrop	−0.6	0.2	−2.7	0.007
Push–pull	−2.4	0.1	−19.0	<0.001
**Fertilizer application**	0.7	0.1	11.4	<0.001
**Maize variety**				
DHO4	0.6	0.2	3.9	<0.001
DK777	−0.1	0.1	−0.6	0.519
DK3081	0.6	0.2	2.7	0.007
Duma43	−0.6	0.2	−3.4	0.001
HB600 series	−0.9	0.1	−8.0	<0.001
OPVs	−0.6	0.1	−6.2	<0.001
PH4	−0.9	0.1	−8.1	<0.001
Pioneer	−1.4	0.1	−9.6	<0.001
**Insecticide use**	1.1	0.1	20.6	<0.001
**Plot size**	0.1	0.0	2.4	0.016
**Previous crop**				
Fallow	−1.7	0.2	−7.2	<0.001
Legumes	0.3	0.1	2.5	0.015
Maize and cassava	17.0	333.6	0.1	0.959
Potatoes	1.4	0.2	6.2	<0.001
Vegetables	0.6	0.2	3.8	<0.001
**Production system**				
Irrigation	2.3	0.2	10.5	<0.001
**Soil types**				
Ferrosols	−5.2	0.5	−10.2	<0.001
Luvisols	−3.4	0.6	−5.7	<0.001
Nitisols	−3.4	0.5	−6.6	<0.001
Vertisols	−4.0	0.5	−7.8	<0.001
**Tillage**				
Conservation tillage	−0.8	0.1	−7.2	<0.001
**Weeding**	1.3	0.1	14.7	<0.001

Note: Monoculture, arenosol (sandy) soils, maize as a previous crop, rainfed as a production system, continuous cropping as cropping pattern, conventional tillage, vegetative crop stage, HB500 series as maize variety, no fertilizer application, infrequent weeding and no insecticide use were reference variables.

**Table 4 insects-13-00266-t004:** Generalized linear models’ results for the disaggregated data on maize field characteristics and crop management practices employed by smallholder farmers interviewed during the survey for fall armyworm infestation and damage in three agro-ecological zones of Kenya in 2019/2020.

Term	Incidence of Plants Infested by FAW											
	Lowlands				Midlands				Highlands			
	Estimate	Standard Error	Z Value	*p* Value	Estimate	Standard Error	Z Value	*p* Value	Estimate	Standard Error	Z Value	*p* Value
**Crop stage**												
Tasselling	−0.9	0.5	−1.8	0.076	−1.5	0.1	−11.8	<0.001	−0.5	0.1	−3.9	<0.001
Mature	−2.0	0.3	−5.9	<0.001	−0.7	0.1	−5.7	<0.001	−0.3	0.1	−2.0	0.042
**Cropping pattern**												
Fallow	−0.7	0.3	−2.3	0.022	−1.4	1.1	−1.3	0.209	−0.9	0.2	−4.0	<0.001
Rotation with legumes	−0.7	0.4	−1.9	0.062	−0.7	1.4	−0.5	0.622	0.7	0.2	4.4	<0.001
Rotation with potatoes	-	-	-	-	−1.1	1.2	−1.0	0.336	0.5	0.2	2.5	0.014
Rotation with vegetables	-	-	-	-	−1.0	1.1	−0.9	0.375	−1.7	0.2	−7.1	<0.001
**Cropping system**												
Maize-cassava intercrop	-	-	-	-	−17.3	350.5	0.0	0.961	-	-	-	-
Maize-legume intercrop	-	-	-	-	−0.8	0.1	−6.2	<0.001	0.1	0.2	0.6	0.552
Maize-legume-cassava intercrop	2.2	1.0	2.2	0.032	−0.7	0.2	−3.3	0.001	-	-	-	-
Mixed cropping (No pattern)	18.5	4269.9	0.0	0.997	−1.9	0.2	−10.6	<0.001	−0.8	0.1	−7.3	<0.001
Maize-vegetable intercrop	−1.9	0.4	−5.0	<0.001	−1.4	0.3	−4.3	<0.001	−0.5	0.3	−1.8	0.072
Push–pull	−2.7	0.4	−7.7	<0.001	-	-	-	-	−2.0	0.2	−13.2	<0.001
Maize-agroforestry	18.5	4269.9	0.0	0.997	-	-	-	-	-	-	-	-
**Fertilizer application**	−0.4	0.3	−1.5	0.144	2.2	0.1	16.0	<0.001	0.6	0.1	5.9	<0.001
**Maize variety**												
DHO4	-	-	-	-	−4.3	0.7	−5.9	<0.001	-	-	-	-
DK777	-	-	-	-	-	-	-	-	0.4	0.2	2.1	0.032
DK3081	−0.6	0.6	−0.9	0.383	-	-	-	-	-	-	-	-
Duma43	-	-	-	-	-	-	-	-	−0.1	0.2	−0.6	0.564
HB600 series	-	-	-	-	−1.1	0.2	−6.9	<0.001	−0.7	0.2	−3.8	<0.001
OPVs	−0.7	0.5	−1.5	0.133	−2.3	0.2	−12.5	<0.001	0.2	0.3	0.9	0.352
PH4	−2.3	0.4	−5.2	<0.001	−1.4	0.2	−12.5	<0.001	−0.2	0.5	−0.4	<0.001
Pioneer	-	-	-	-	−2.0	0.3	−7.9	<0.001	−0.8	0.2	−3.8	<0.001
Panner	-	-	-	-	−5.0	1.0	−4.9	<0.001	−2.8	0.6	−4.4	<0.001
**Insecticide use**	−0.9	0.5	−1.8	0.078	1.8	0.1	16.1	<0.001	−1.4	0.3	−4.9	<0.001
**Plot size**	0.0	0.1	−0.4	0.690	−1.0	0.1	−13.2	<0.001	0.9	0.1	14.4	<0.001
**Previous crop**												
Fallow	-	-	-	-	-	-	-	-	−1.0	0.2	−5.1	<0.001
Beans	−0.3	0.5	−0.6	0.541	1.2	1.1	1.1	0.266	−1.1	0.2	−4.5	<0.001
Potatoes	-	-	-	-	0.1	0.7	0.1	0.883	1.5	0.3	5.8	<0.001
Tomatoes	-	-	-	-	−0.6	0.7	−0.8	0.411	-	-	-	-
Cabbage	-	-	-	-	1.2	1.1	1.1	0.266	-	-	-	-
**Production system**												
Irrigation	2.3	0.2	10.5	<0.001	0.2	0.7	0.3	0.779	-	-	-	-
**Soil types**												
Luvisols	−3.4	0.6	−5.7	<0.001	-	-	-	-	-	-	-	-
Nitisols	−2.1	0.5	−4.0	<0.001	−0.3	1.1	−0.2	0.809	1.0	0.1	8.6	<0.001
Vertisols	-	-	-	-	−0.4	1.2	−0.4	0.722	0.9	0.2	5.7	<0.001
**Tillage**												
Conservation tillage	−1.3	0.3	−4.5	<0.001	0.6	0.6	1.0	0.332	−0.9	0.1	−8.6	<0.001
**Weeding**	−0.3	0.3	−1.0	0.306	−0.8	0.6	−1.4	0.171	1.0	0.1	8.0	<0.001

Note: - Indicates absence of the assessed parameter. Monoculture, arenosol (sandy) soils (for lowlands), ferrosol soils (for midlands and highlands) maize as a previous crop, rainfed as a production system, continuous cropping as cropping pattern, conventional tillage, vegetative crop stage, HB500 series as maize variety, no fertilizer application, no weeding and no insecticide use were reference variables.

**Table 5 insects-13-00266-t005:** Analysis of variance results for the aggregated data on maize field characteristics and crop management practices employed by smallholder farmers interviewed during the survey for fall armyworm infestation and damage in three agro-ecological zones of Kenya in 2019/2020.

Term	Damage Score from the Davis Scale				
	Degrees of Freedom	Sum of Squares	Mean Squares	*F* Value	*p-*Value
Crop stage	2	9.0	4.5	0.7	0.514
Cropping pattern	4	22.7	4.5	0.7	0.609
Cropping system	6	221.2	31.6	9.7	<0.001
Fertilizer application	1	13.1	6.6	1.0	0.377
Maize variety	8	122.2	17.5	3.8	0.003
Insecticide use	1	100.2	100.2	20.1	<0.001
Plot size	3	17.0	4.2	0.6	0.646
Previous crop	5	64.3	12.9	2.2	0.069
Production system	1	28.2	28.2	4.5	0.038
Soil types	4	144.3	36.1	8.1	<0.001
Tillage	1	18.0	9.0	1.4	0.260
Weeding	1	45.8	45.8	7.7	0.007

**Table 6 insects-13-00266-t006:** Analysis of variance results for the disaggregated data on maize field characteristics and crop management practices employed by smallholder farmers interviewed during the survey for fall armyworm infestation and damage in three agro-ecological zones of Kenya in 2019/2020.

Term	Damage Score from the Davis Scale														
	Lowlands					Midlands					Highlands				
	Degrees of Freedom	Sum of Squares	Mean Squares	*F* Value	*p-*Value	Degrees of Freedom	Sum of Squares	Mean Squares	*F* Value	*p-*Value	Degrees of Freedom	Sum of Squares	Mean Squares	*F* Value	*p-*Value
Crop stage	2	17.3	8.6	4.0	0.038	2	14.0	7.0	4.5	0.056	1	0.4	0.4	0.1	0.747
Cropping pattern	2	16.8	8.4	3.8	0.043	4	4.5	1.1	0.3	0.882	6	32.9	5.5	2.0	0.115
Cropping system	1	6.6	6.6	2.4	0.141	2	1.4	0.7	0.2	0.822	5	59.2	19.7	13.5	<0.001
Fertilizer application	1	3.9	3.9	1.4	0.254	1	1.8	1.8	0.6	0.458	1	8.0	8.0	2.5	0.124
Maize variety	3	33.0	11.0	13.4	<0.001	3	0.3	0.1	0.0	0.995	6	18.5	3.1	0.7	0.622
Insecticide use	1	9.0	9.0	3.6	0.074	1	3.5	3.5	1.3	0.287	1	9.7	9.7	3.1	0.089
Plot size	2	18.6	9.3	4.4	0.028	1	0.3	0.3	0.1	0.781	2	46.6	23.3	12.4	<0.001
Previous crop	1	1.1	1.1	0.6	0.463	2	0.5	0.3	0.1	0.868	3	21.0	7.0	2.5	0.082
Production system	1	0.2	0.2	0.1	0.800	1	1.1	1.1	0.4	0.563	1	30.1	30.1	12.6	0.001
Soil types	2	12.5	12.4	5.1	0.037	2	10.1	5.1	2.4	0.163	1	8.6	4.3	1.3	0.287
Tillage	1	0.2	0.2	0.1	0.807	1	6.4	6.4	2.8	0.136	1	11.4	11.4	3.7	0.064
Weeding	1	1.2	1.2	0.4	0.539	1	4.5	4.5	1.8	0.222	1	10.7	10.7	3.5	0.073

## Data Availability

Data can be provided on request from the lead author.

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
