# Peer review of "Agronomic Factors Influencing Fall Armyworm (Spodoptera frugiperda) Infestation and Damage and Its Co-Occurrence with Stemborers in Maize Cropping Systems in Kenya"

_insects, 2022, doi:10.3390/insects13030266_

Round 1
Reviewer 1 Report
Review comments
I commend the authors for putting up this piece of work that sought to understand the extent of infestation and damage of FAW/stemborers on maize in different agro-ecological zones and under different management practices. The work is novel and builds on the existing body of scientific literature of fall armyworm in Africa.
While the work is good, there is room for improvement as indicated in the suggestions below:
- Tables
- The table titles should be informative and depict content. For instance, Table title is not so informative. It omits the plot size information, which was not measured in percentage. Align plot variable to the left
- Big tables like 1, 4 and 6 could be presented as supplementary tables
- Table column headings and naming of variables must be consistent in all tables in the main text and supplementary tables. For instance, Table 1 has fertilizer and insecticide use denoted as “Yes” and “No” and the supp Tables have “insecticide use” and “no insecticide use”
- Figures too should have informative figure legends, which can make the figure stand alone. Also figure axes units must be consistent with the legend content. For instance, Figs 4 and 5 have Y axes indicating that the means are for 50 plants, and yet the legend indicates per plant
- Infestation, damage and other parameters should be reduced to one decimal point, the SE can remain as is
- Check the text for conformity with space between numbers and standard units, e.g Lines 143 – 1200 mm; Line 148, 500 – 1000 mm. Line 144, check the symbol for degrees; Lime 261 – 0.56 ha
Methods
This section should be improved:
- All the data collected in this section should be analyzed, presented and discussed.
- Data on types and dosage of insecticides/herbicides were collected but not used anywhere and yet they could help with the discussions
- Line 217, infestation data was not collected, but rather derived from severity/damage data
- There is a need to explain and justify how stemborer data were collected, and why data were not collected on the species of stemborers in the different agro-ecological zones.
- There are known differences on the occurrence of different stemborer species in different agro-ecologies and at different phenological stages of the maize crop, and these could affect the interpretation of the results.
- Furthermore, while it is evident that FAW has displaced stemborers, sampling for stemborer larvae only in the whorls gives an underestimate of the larval and pupal population since most second and later instars enter the maize stems, and not only limited to the whorl area.
- The objective on co-occurrence targeted ascertaining the magnitude of co-occurrence. The magnitude question is not answered. Moreover, one would expect data on fields/plants with FAW alone, FAW + stemborers, and stemborers alone in order to answer such a question.
- The Davis Scale was developed for screening whorl-stage of maize, and yet the same scale was used for collecting damage data on tassels, silks and ears. How was it adapted for the latter stages, and how were the data for different stages combined for analysis
- There is no mention on how were soil characteristics/type data collected
Results
- You could swap sections 3.1 and 3.2 so that the respondents characteristics come before field characteristics
- Line 189 – should it be below 35 or 36 years?
Discussions
- I have a feeling the discussion section is lengthy and could be summarized.
- Line 401/402 – How did the authors arrive at the conclusion that maize was predominantly grown for subsistence
- The authors have discussed findings, but failed to acknowledge limitations of the study such as unequal numbers of counties sampled per zone and lumping of all stemborer species as one
Conclusions
The conclusions in simple summary, abstract and conclusions sections should be consistent and answer the key words in the objectives: Objective 1 infestation and damage across AEZs and agronomic practices; Objective 2 – Agronomic factors and ; Objective 3 – Magnitude of co-occurrence
Author Response
Thank you for your comments and suggestions. We have addressed them in the attached document.
Reviewer 2 Report
Insects-1575952
Title: Agronomic factors influencing fall armyworm infestation and damage and its co-occurrence with stemborers in maize cropping systems in Kenya
Brief.
This manuscript conducts a large-scale survey of 180 farms, including field survey and questionnaire survey. And it takes twice as long (23 pages) to describe it. However, authors devote so much text to the results and discussion sections that there are too few lines that are closely related to the topic. The results section, 3.1 and 3.2 state the general characteristics of the farm and farmers, which is completely irrelevant to the subject. 3.3 and 3.4 are the most important parts of the article, but one is FAW Infestation and the other is FAW damage. The content of the two parts is similar, but just with different algorithms. It is suggested to use the most accurate one. The 3.5 is only a comparison of FAW and corn borer abundance. In the discussion section, the authors express many things that are not related to the results. For example, in the four paragraphs L482-535, the authors respectively devote a whole paragraph to discussing natural enemies, plant characteristics, irrigation system and insecticides, which are not represented in the results. This makes the discussion section look more like a review article. It is suggested to delete the sentences irrelevant to the topic in the full text, otherwise the article will be difficult to grasp the main point. So I would advice the authors to revise those issues and resubmit later.
General comments
L33-34 The authors are advised not to mention the “coastal” here. The authors mainly illustrate the influence of altitude on FAW distribution. But the inclusion of the word "coastal" is misleading, because coastal areas are often saline and alkaline, and this is also an important factor in the FAW density, which is easy to misunderstand.
L38-39 The authors need further clarification. It is well known that different maize varieties lead to different FAW effects. Readers are most concerned about which or which varieties are least affected by FAW. Here the author needs to make it clear.
L39-41 I'm not sure which of the above results show that FAW is superior to the stemborer. It was the only sentence that seemed to matter: “ Its population (eggs and larvae) dominated that of resident stemborers in maize fields (L34-35)”. While other results are described by different factors affecting FAW infestation. The authors are advised to revise this sentence because the results above do not indicate this conclusion.
L129 The authors describe the features of the study site in great detail, but suggest adding the coordinates and altitudes of 180 sites to show the situation of the study site more clearly.
L167 A more detailed description is needed here, did each farm have the same amount of corn or acreage surveyed, how many corn plants were surveyed on each farm, or how many acreage of corn fields were surveyed, and how was the survey conducted, whether it was observation or dissection of corn plants.
L209 How was systematic sampling done here, resampling from 180 or was it systematic sampling that ended up with 180 samples.
L213 Why the authors did not record the number of adults, which makes the count of abundance somewhat skewed. FAW adults can be collected with a vacuum pump or, if conditions are limited, recorded in the form of observation, but adults must be included in the statistics.
L259 This paragraph is not appropriate in the results section, it is not closely related to the topic. There is no related description statement in the result part of abstract, which further indicates that it does not contribute much to the topic. A discussion or attachment is more appropriate.
L284 Again, this is not very relevant. If the authors want to keep it in the results section, they suggest adding "Characteristics of farms and farmers" to the title and adding a description in the abstract section.
L301-302 It can only be seen from the results in Figure 2 that FAW is more effective in low-altitude areas than in high-altitude areas. There is no significant difference between low-altitude and middle-altitude areas.
L305-309 As I mentioned earlier, maxima and minima need to be indicated so that readers can quickly grasp what the authors are trying to convey.
L347 I don't know why the authors have redone FAW infestation. It seems to be just a new calculation method, and there is no comparison between the two methods. I suggest choosing the most accurate method in the text.
L384 It is recommended that Figure 4 and Figure 5 be merged.
L400 The discussion section should be completely rewritten. The authors make a lot of statements in this section, but many are not relevant to the topic, this section looks more like a review. For example, the authors used one paragraph in each of these four paragraphs (L482-535) to describe natural enemies, plant characteristics, irrigation systems, and pesticides, but these are not closely related to your results. These can be included in the discussion section, but in the form of a full paragraph discussion they confuse readers. The discussion is not well focused due to too many words, it is advised to simplify statements.
Author Response
Thank you for your useful comments. We have addressed them which has greatly improved the manuscript.

Reviewer 3 Report
This manuscript describes a study carried out in Kenya to assess, through farmers’s surveys and direct observations on farm, which agronomic factors influence infestation levels of fall armyworm (FAW) and their damage to maize. FAW is a very important invasive moth in Africa and, to develop sustainable control methods, in particularly through agroecological approaches, there is an urgent need to better understand which farming practices and environmental conditions favour or affect FAW infestations and damage to crops. Thus, this study is clearly of high interest. The study also describes the co-occurrence of FAW and stemborers, which were previously the main insect pests in maize cropping systems. This part is less interesting because does not provide data on their “interactions” nor on the mechanisms of interactions. I don’t think that this part is necessary, but it does not harm the paper and, thus, it could also stay.
The manuscript has been written and prepared with care. I also enjoyed reading the discussion, which shows that the authors understand the topic very well.
I have only one general concern. There is a very high number of independent variables in the analyses on FAW infestation and damage (not to speak of all the factors related to the farm that were not entered into the analyses) and, for this number of factors, relatively few fields were involved (180). Despite the statistics used, confounding factors are very difficult to avoid. Considering that the agroecological zones vary not only in climate conditions but most probably in many other aspects related to agricultural practices, economy, etc., it is good that the authors also analysed the data disaggregated for each zone. However, then the number of replicates is even lower (e.g. 20 fields in the lowlands).
To say it differently, to study the effect of factors such as intercropping or tillage on FAW infestation and damage, it would probably be better to observe 180 fields in the same area at the same time, minimising the effect of other factors (e.g. use only unsprayed fields, similar seeds, fertilization, etc.)
Here are some specific concerns/questions regarding the data and the analyses:
L.210-212. The period during which surveys and data collection were organised is quite long and variable. Was this organised to conduct data collections in the different regions in a similar moment in the cropping seasons? This question came to me especially when I saw, in Table 1, that 80% of the fields were irrigated in the Midlands and only 5% in the 2 other zones. This suggests to me that the observations were made off-season in the Midlands and in the rainy seasons in the two other zones (but I may be wrong). In such biased cases, how can one really assess the effect of irrigation on FAW?
In Table 4, indicate what does “-“ mean.
In table 4, fertilizer application. In Midlands and Highlands, the sign of Z is positive, suggesting that fertilization increases FAW abundance. However, in L. 329 it says that fertilizer application “recorded low FAW infestations”. Isn’t there an error somewhere?
In Table 4, Insecticide use seems to have opposite effects in the Midlands and in the Highlands. But, again, in L. 329, the text does not suggest that.
Regarding insecticide use and the discussion section on insecticides starting L. 522, it seems that the first sentence contradicts the results that show significant differences in insecticide use, albeit in different directions in the different zones. However, it is almost impossible to assess the effect of insecticides in such studies because insecticides are often applied when there are signs of damage, i.e. FAW infestation and damage level are not necessarily the result of the insecticide application but the application may result from high infestation and/or damage. For example, in the Highlands, it is the low level of FAW infestation (probably due to climate) that generates the low level of insecticide use, not the contrary. The authors could moderate their discussion, which suggests that insecticides have no effect on FAW. Many studies showed that it has an immediate effect, even though, at landscape scale, the long term effect of heavy insecticide on natural FAW control is probably negative.
And some minor comments on the text:
-All scientific names should be put in italics
L.18 and L 33. Isn’t it more correct to say “in costal lowlands”?
L.57. Include Latin name and authority for FAW. This is the first citation. Abstracts don’t count.
L.165. Delete “of”?
L. 323. difference instead of different
L. 439. The degree sign ° is wrongly placed twice.
L. 576. Parasitoids and predators?
L. 613. systems instead of systemS
Author Response
Thank you for your comments and suggestions. We have addressed them which has greatly helped improve the manuscript.

Round 2
Reviewer 1 Report
Good improvement has been made. However, it can still be improved further. Details are specified below.
Check section numberings in methods and results
Table heading and figures could take the broad categorization (Farm characteristics, cropping system, patterns and crop management practices) stated in sections 3.1, 3.2, 3.3 etc. Authors could adapt the suggestions below. Details can be provided as footnotes
Round off to one decimal point was only effected in the tables and not the data provided in results and discussion sections.
Table 1
Socio demographic characteristics of smallholder farmers interviewed during the survey for fall armyworm infestation and damage in three agro-ecological zones of Kenya, year
Table 2:
Maize field characteristics and crop management practices employed by smallholder farmers interviewed during the survey for fall armyworm infestation and damage in three agro-ecological zones of Kenya, year
Table 3
Generalized linear models' results for the aggregated data on maize field characteristics and crop management practices employed by smallholder farmers interviewed during the survey for fall armyworm infestation and damage in three agro-ecological zones of Kenya, year
Table 4 – Revise as suggested above
Table 5 – borrow from Table 3
Figure 2 legend – replace “across” with “in” …..different agro-ecological zones of Kenya
Figure 3
Mean fall armyworm damage score (± SE) on leaves of maize plants in three different agro-ecological zones of Kenya.
Y axis = Mean leaf damage score
Figure 4
Percentage infestation……………………….
Line 10, delete “that”
Line 13, transfer across different agro-ecological zones to the end of the sentence and ensure flow with “in different maize cropping systems”. This should be consistent with text in Line lines 24/25
Lines 42 and 43, is it true that maize is the third most important crop for food and cash?
Line 50, the authors should again review the losses in reference Number 4 (page 708) and quote appropriately. Kfir et al reviewed losses in different countries and for different stem borer species under different conditions. The 80% loss is quoted for C. partellus in sorghum in Kenya
Line 53 is a wrong attribution to Prassana et al 2018.
Line 54, the first detection of fall armwyorm in Africa was reported by Goergen et al 2016, and not by De Groote et
Line 75 change rate to level. This apples to the entire document where rate is used.
Line 80, some respondents first observed FAW in 2015 and not 2016 as reported.
Line 123 Sea mays should come earlier at the first mention
Section 2.1 from Line 114. The categorization of zones into low, mid and high altitudes is not clear. Low altitude ranged from 0 – 900 m asl, mid from 900 to 1800, and high up to 2700 m asl.
Does this mean an area of 900 m asl was considered low of mid latitude depending on the zone it was sampled from?
What was the lower level of high altitude?
Line 154, delete “undertake active field sampling”
Line 155, delete thereafter
Line 184, were ox ploughs and tractors the only preparation equipment in all the conventional farms?
Line 200, omit tassels, silk and ears
Line 204, specify whether calculation of average damage score included or excluded apparently undamaged plants
Section 3.1, round of figurers to one decimal point
Line 224, range should be stated as “from xxxxx to”
Line 238, delete on average
Section 3.2, round figures to one decimal point
Line 333 – 334, move text to analysis section.
The reason for excluding the species level identification of stem borers is not convincing, given that (Lines 349 to 350) states that, “FAW and stem borers were found in most of the surveyed fields”. Moreover, if your interest was in Sesamia too, I believe that it is a borer species that attacks towards maturity.
If calculation of infestation involved assessment of damage caused by each pest species, what were the distinctive damage features for each species, especially at the vegetative stage?
Section 3.5. Look at prevalence, infestation level and abundance. Abundance data were not collected
Sentence in Lines 445 to 446 (ref 33) is redundant, delete and add reference to the preceding sentence
Line 449, does rainfall wash off or suffocate FAW pupae?
Line 498 to 502, so maize is the most preferred host plant, what prospects are there for a more attractive host plant than maize?
Conclusion
Has answered the question on prevalence. Infestation and damage in different agro-ecological zones is not answered
It would ne good to state the direction of influence
What is the magnitude of occurrence and what was it before?
Lines 540 to 542, important to recommend incorporation for those tested practices, and evaluation and or validation of others
Author contributions
Check the initials of the authors against the names. Should Andre’s be PC or PAC?
Agreed upon criteria such as resources, data curation, visualization, supervision and project management are omitted
References
Check all references in the reference for completeness and consistency in formatting and writing article titles, scientific names etc. E.g. reference 28, 30, 31,
Scientific names in italics
Line 588, spelling of Spodoptera
Line 590, space before, “management”
Line 597, 600, check genus name
Letter 598, 638, armyworm with lower case, “a”
Reviewer 2 Report
insects-1575952 R1
Title:
Agronomic factors influencing fall armyworm (Spodoptera frugiperda) infestation and damage and its co-occurrence with stemborers in maize cropping systems in Kenya
The manuscript has been greatly improved, but there are still some issues that need to be revised, mainly in the discussion section. The presentation format needs to be consistent, fall armyworm is used in some places and FAW is used in others.
L375-378 In the results section, the authors used 3.1 and 3.2 to present the data, that's a big part. But the authors discuss these data in just three sentences. It is suggested that the authors further explore these data.
L379-407 These two paragraphs both express that FAW affects low altitude areas more than middle and high altitude areas. It is suggested to combine these two paragraphs.
L426 It is appropriate that the authors put the natural enemy statement here and suggest providing several examples of parasitoids and predators.
L446-453 This paragraph is mainly about the irrigation system, the rainfall is not related to the irrigation system, it belongs to the climate category, it is suggested to delete this part.
L457-459 I wonder why pesticide application has no effect on FAW damage, or even higher effect in some areas. Suggest an explanation.
L491 In practice, weeding is less economical than intercropping as mentioned earlier by the authors.
